# Rehabilitation Outcomes Following Hip Fracture of Home-Based Exercise Interventions Using a Wearable Device—A Randomized Controlled Pilot and Feasibility Study

**DOI:** 10.3390/ijerph20043107

**Published:** 2023-02-10

**Authors:** David Flecks Howell, Agneta Malmgren Fänge, Cecilia Rogmark, Eva Ekvall Hansson

**Affiliations:** 1Træningscenter Brøndby, Horsedammen 36A, 2605 Brøndby, Denmark; 2Department of Health Sciences, Faculty of Medicine, Lund University, P.O. Box 157, 22100 Lund, Sweden; 3Department of Orthopedics, Skane University Hospital, Lund University, 21428 Malmö, Sweden

**Keywords:** balance, hip fracture, HIFE, rehabilitation, wearable device, postural sway

## Abstract

Although hip fractures are common and severe, there is insufficient evidence concerning which type of rehabilitation is most beneficial. The primary aim of this three-armed pilot study was to investigate any difference in outcome after hip fractures between and within groups in terms of balance, everyday activities, and health-related quality of life (HRQoL) following different home rehabilitation interventions. Further aims were to study feasibility and to suggest, if necessary, adjustments to the protocol for a future full randomized controlled trial (RCT). In total, 32 persons were included in this study. The intervention groups underwent the HIFE program with or without an inertial measurement unit, while the control group underwent standard rehabilitation. Within- and between-groups differences in outcomes and feasibility outcomes in terms of recruitment and retention rates were analyzed, and the ability to collect primary and secondary outcomes was assessed. Balance, measured as postural sway, showed no significant improvement in any group. All three groups improved in functional balance (*p* = 0.011–0.028), activity of daily living (*p* = 0.012–0.027), and in HRQoL (*p* = 0.017–0.028). There were no other significant changes within or between the groups. The recruitment rate was 46%, the retention rate was 75%, and the ability to collect outcome measures was 80% at baseline and 64% at follow-up. Based on the results, it is possible to, after adjusting the protocol, conduct a full RCT.

## 1. Introduction

By 2050, around 8 million hip fractures are expected to occur per year [1,2]. This will put a burden on health care systems, as more individuals will need rehabilitation in order to regain independence [3].

A hip fracture increases a person’s fear of falling and, thus, restricts activity [4]. Moreover, the risk of a new fracture within the first 2 years is high, and effective and early rehabilitation is therefore important [5]. Rehabilitation after a hip fracture should conform to the patient’s needs and level of mobility, and rehabilitation plans should focus on balance, functional activities, and endurance and progressive strength training for the lower limbs [6]. Many of the risk factors for hip fractures [7], such as falls, physical inactivity, muscle weakness, chronic health conditions, impaired cognition, and impaired vision, also affect the capacity to participate in rehabilitation [6]. These factors vary between individuals and emphasize the need for individual rehabilitation plans [8], especially for frail older adults [6]. Even if there is evidence concerning the structure of rehabilitation plans [6], factors such as training at a too-low intensity, possibly resulting in a less effective outcome, has been identified [9]. A study protocol has been published where we address the importance of interventions that prevent fractures and the lack of high-quality research on hip fractures [10]. Moreover, what type of exercise that is most beneficial for this population is not fully explored [11,12]. A lack of high-quality RCTs’ exploring the effects of rehabilitation after hip fractures among older people has also been identified [13].

One type of rehabilitation program is the High-Intensity Functional Exercise (HIFE) program, which contains exercises for balance, progressive strength training, and functional exercises. The HIFE program was developed for frail older adults with lower limb problems at different functional levels living in a nursing home. It is designed to help them improve balance, lower-limb strength, and mobility in their own home [14]. Using an individualized program, such as the HIFE program, might make it easier to tailor individual rehabilitation programs. Moreover, the HIFE program also gives the therapist the ability to monitor the intensity and the type of exercise that is used. Compared to standard rehabilitation, this ensures progression in rehabilitation. To further reach individual rehabilitation goals, a wearable device that measures the number of daily steps, stride length, gait flexibility, and postural sway can be used. Since gait speed is a predictor of mortality [12] and gait speed, step length, and postural sway are related to fall risk [11], these measures are relevant outcomes of rehabilitation programs. Moreover, general physical activity has a strong correlation with general health [15], and physical activity is therefore also relevant as an outcome measure of rehabilitation programs. With new and wearable technology [16], factors such as physical activity, daily steps, and gait speed, can be measured and monitored 24 h a day.

In the Medical Research Council (MRC) framework for designing and evaluating complex interventions in health care, feasibility studies constitute important steps to inform further full-scale testing, e.g., by performing an RCT [17]. Furthermore, feasibility measures, such as retention and recruitment rates, are important in order to test the planned processes of a trial [18].

The primary aim of this pilot and feasibility study was to investigate the feasibility of an RCT in terms of the rates of recruitment, retention, and adverse events, as well as regarding outcome data collection. A further aim was to study outcome differences after hip fractures regarding balance, everyday activities, and health-related quality of life across groups following different home rehabilitation interventions. Furthermore, if found necessary, suggested adjustments to the protocol could be made to conduct a full randomized controlled trial (RCT).

## 2. Materials and Methods

### 2.1. Study Design

A three-armed pilot study was designed. Two intervention groups were included, namely, (1) participants following the HIFE program and (2) participants following the HIFE program, with the addition of data collection using a wearable device measuring body position and movement, and (3) a control group. The design of this study has been published in a detailed study protocol [10]. The study was registered in Clinical Trials (NCT04906265).

### 2.2. Study Setting

The study was performed in Malmö, the third largest city in Sweden, with approximately 330,000 inhabitants. Every year, about 700 individuals are admitted and treated for a hip fracture at Skåne University Hospital in Malmö [3].

### 2.3. Recruitment and Participant Inclusion/Exclusion

The inclusion criteria were being hospitalized, treated for a hip fracture, and in need of home-based rehabilitation. The exclusion criteria were (1) not being able to read and understand Swedish, (2) having major neurological diseases impacting walking and balance, and (3) having a diagnosed cognitive disease or severe cognitive impairment affecting the ability to perform exercises or use the wearable device, which was assessed by a physiotherapist.

Those fulfilling the inclusion criteria were informed about the study after hospital discharge by a physiotherapist (PT). Written informed consent was provided by those agreeing to participate.

### 2.4. Randomization

A cluster randomization procedure was applied, a 6-based cluster, thus giving each cluster 2 participants to each intervention arm. Sealed envelopes were used. The numbers were randomly drawn by a person that had no contact with the participants or the researcher.

### 2.5. Intervention

All participants received an individually tailored intervention in either one of the intervention groups (HIFE) or the control group (standard rehabilitation). In all three groups, home visits by the physiotherapist were included. At these home visits, exercises were added or adjusted when needed.

Intervention group 1: This group followed the HIFE program, and other exercises were added when needed. The HIFE program consists of 5 training categories depending on the individual’s level. In every category, several exercises are included (Table 1). To select exercises, a task of walking a short distance (5–10 m) without any walking aid guided the PT in selecting exercises for the individual. Based on the repetition maximum (RM) and the ability to adequately maintain postural stability though challenged, the intensity of the strength and balance exercises was chosen by the PT (Table 2). Before starting the exercise, a 5 min warm up consisting of 8 movements was performed: walking on the spot, reciprocal arm swings, raising and lowering the arms in various directions, knee stretches, and alternately taking steps sideways and backwards with each leg.

Intervention group 2: This group followed the HIFE program but was also given feedback by an inertial measurement unit (IMU), which they wore 24 h a day until the end of the rehabilitation period (Figure 1). The PT could then give specific information to the individual about the progress of the rehabilitation. The IMU was mounted by the PT on the participant’s right thigh under the clothes, approximately 10 cm above the knee, with skin-friendly adhesives. The data from the IMU were sent through a router to a platform through which the PT could obtain the data and follow the participants’ progress in the rehabilitation. The data collected when the participant was too far away from the router were stored in the IMU and were uploaded when the IMU was recharged (twice a week). The IMU used was the “Snubblometer” 2022, version 1.0, from Infonomy (www.infonomy.com) (Figure 1). The IMU measures body position (time spent supine, prone, side lying, sitting, standing, and walking) and movement (strides, step length, variation in gait, falls, and near falls). With a wearable device, the therapist receives feedback and can ensure that the rehabilitation program is tailored to the needs of the individual [8]. IMUs are valid and reliable for measuring outcomes in persons at risk of falling [16,19,20,21]. For this study, hardware and algorithms for deciding step length, step time, and walking speed were used, which are identical to those used in earlier validation projects, where they showed a correlation of 0.96–1.00 for step time and 0.83–0.93 for step length in studies on validity and an intraclass correlation of 0.83–0.89 for reliability [22,23].

The equipment used in HIFE was step boards, chair cushions (5 cm height minimum), weighted belts (1 kg), soft pads, mattresses, balls, bean bags, belts with handles, and chairs without arm support; all equipment was portable. Progression for each exercise could, for example, be achieved by providing less assistance or softer surfaces or by increasing the load of the weight belt. 

Control group: Participants were offered individually tailored standard rehabilitation, which included walking and functional exercises [12]. The intervention was tailored to each participant according to the type of fracture, the type of surgery, an assessment by the PT, and the treatment goal.

### 2.6. Outcomes

Feasibility was the main outcome measure and measured by means of *recruitment rate*, calculated as the percentage of individuals who agreed to be in the study among those who were qualified; *retention rate*, calculated as the percentage of individuals who completed the intervention; *the ability to collect data*, calculated as the percentage of completed collected data at baseline and follow-up; and *adverse events*, which is the number of harmful or negative events that may have influenced the study procedure. It was decided that, if the majority of feasibility measures was rated as low and there was no possibilities to adjust the protocol in order to raise the measures from low to acceptable or high, the full RCT would not be performed.

Patient-related outcomes were balance-measured as postural sway, functional balance, functional independence in everyday activities, and health-related quality of life (HRQoL). Measures of postural sway, especially in the medio-lateral direction, have been shown to be indicators of future falls [24]. Research shows that a larger postural sway independently increases the risk of future falls by 75% when the statistical model is adjusted for multiple confounders [25]. Postural sway was measured with an IMU placed at a level of L5 and with feet 10 cm apart at the heels and toes pointing 30° outwards for 30 s with eyes open. The IMU has shown good validity and reliability for measuring postural sway [16]. Functional balance was measured using the Functional Balance Test for Geriatric Patients (FBG). The FBG evaluates an individual’s mobility in 4 activities: to sit and stand up, maintain standing position, walk, and turn. Every activity has 7 levels of difficulty indicating the level of independence. The individual is scored from 0 (no execution) to 6 (a higher level of performance). The maximum score is 24. The FBG has shown good validity and reliability compared to the Bergs Balance Scale, BBS [26]. Functional independence in everyday activities was assessed using the Barthel Index of Activity of Daily Living Questionnaire (BI). BI consists of 10 everyday activities, and individuals are asked how well they manage each activity independently. The answers are scored on a scale from 0 to 2, where 0 indicates the least independence and 2 indicates the highest independence. The score is then multiplied by 5 to achieve a score on a 100-point scale; i.e., the score ranges from 0 to 100 [27]. BI has shown good validity and reliability for older outpatients and for those after a hip fracture [28,29].

HRQoL was evaluated using the EQ-5D [30]. It consists of 2 parts: First, an evaluation of 5 dimensions (mobility, self-care, usual activities, pain/discomfort, and anxiety/depression). Every dimension is scored on a 5-point scale (no problem, slight, moderate, severe, and extreme). In the second part, the respondent evaluates their overall health on a Visual Analogue Scale (VAS) [31]. The EQ-5D has shown good validity and reliability [32]. The index value reflects health status compared to that of the general population in a specific country or region. It ranges from 1 (perfect health) to 0 (death) [33]. To calculate the EQ-5D index in this study, the Danish crosswalk index was used [34].

### 2.7. Measurement

An IMU measures linear acceleration and rotation. The study used 2 different types of IMUs, one for intermittent measurements of postural sway and one for continuous measurements of body positions and movement. The IMU that measured postural sway was used in all 3 groups to measure postural sway in the anterior/posterior and medio/lateral directions at baseline and follow-up. During the measurements, the IMU was attached to the back, and the participant stood still for 30 s. An IMU attached to the back has shown an equally good ability to discriminate between different conditions in postural sway as other placements [35]. A large postural sway has been used as an indicator of having a high fall risk [36]. Therefore, postural sway is considered a good method of measuring the outcomes of rehabilitation following a hip fracture [36,37].

### 2.8. Procedure

Before the data collection, all health care personnel involved in hip fracture rehabilitation in Malmö municipality were trained in the HIFE program and in using the IMU device. Several workshops were held with the participation of leaders from the rehabilitation unit and technicians who developed the IMU device. After the data collection was completed, a final workshop was held to obtain information from the involved physiotherapists regarding the feasibility of the study.

Data on primary and secondary outcomes were collected at baseline, together with participant characteristics (age, sex, the method of surgery, and medication), by the PT. Data on outcomes were also collected at the end of the rehabilitation period. The responsible PT also took notes in case there were deviations from the protocol or withdrawal from the study.

### 2.9. Statistical Methods

A previous study was used to calculate statistical power [36]. From that study, the significant difference between fallers and non-fallers in anterior/posterior postural sway was 1.8 mm/s, with a standard deviation of 3.6 mm/s for fallers and 2.6 for non-fallers. In order to reach 80% statistical power with a significant difference of <0.05, *n =* 43 in each arm was required for the full RCT. To conduct the pilot study, 20% was needed, which meant 9 individuals in each group.

The data were not normally distributed, and due to the small sample sizes, the difference in the primary and secondary outcomes between the 3 groups was analyzed using the Kruskal–Wallis test. To measure within-group differences, the Mann–Whitney U test was used. No adjustment for the number of comparisons was conducted, as all the variables were analyzed independently. Following the CONSORT statement, on-treatment analyses were applied. SPSS version 28.0 was used (SPSS Inc., Chicago, IL, USA).

## 3. Results

A total of 32 participants were included in the study, 25 women and 7 men aged between 56 and 95 years of age (Md 85 years), during the period of January 2021 to June 2022. A total of 10 persons had had a hip arthroplasty, 19 had had intramedullary nail fixation, and 3 had had other surgical treatments. A total of 22 persons used cardiovascular medication, 4 used psychotropic medication, 2 used anticonvulsants, and 22 used other medication. Pain medication in relation to the hip fracture was used by 24 persons.

Among the 32 persons included in the study, 11 were randomized to intervention group 1 (HIFE program), 10 were randomized to intervention group 2 (HIFE program + IMU), and 11 were randomized to the control group (Figure 2). In total, there were 7 dropouts: 5 for unknown reasons and 2 due to hospitalization. The baseline measures for the total group and for each group are shown in Table 3.

### 3.1. Feasibility Measures

In total, there were 61 qualified participants for the feasibility study. The recruitment rate was 52%, and the retention rate 78% at the primary follow-up assessments. Seven participants dropped out of the study. The reasons for drop out were unknown reasons (*n =* 5) and hospitalization (*n =* 2). At baseline, 80% of the primary and secondary outcomes were collected (*n =* 32) (FBG, BI, EQ5D index, and EQ5D VAS = 100%), (PSOEAP and PSOEML = 68%), (PSCEAP and PSCEML = 53%), and at follow-up, the corresponding figure was 66% (FBG and BI =78%), (EQ5D index and EQ5D VAS = 75%), (PSOEAP = 59%), (PSOEML and PSCEAP = 56%), (PSCEML = 53%) (*n =* 25).

### 3.2. Adverse Events

No harmful events occurred during the trial period. However, five events were observed during the data collection that negatively influenced the study procedure, namely, problems with the IMU device, such as not being used, having a low battery, or dysfunction.

### 3.3. Within-Group Differences

All three groups improved in functional balance (FBG) (intervention group 1 *p* = 0.007, intervention group 2 *p* = 0.018, and control group *p* = 0.018), functional independence in everyday activities (BI) (intervention group 1 *p* = 0.008, intervention group 2 *p* = 0.011, and control group *p* = 0.012), and quality of life (EQ5D index) (intervention group 1 *p* = 0.011, intervention group 2 *p* = 0.012, and control group *p* = 0.018). There were significant changes in intervention group 2 and in the control group for EQ5D VAS (*p* = 0.036 and *p* = 0.018) but not in intervention group 1 (*p* = 0.262). There were no significant differences in postural sway within any group (*p* > 0.05) (Table 4).

### 3.4. Between-Group Differences

There were no differences in changes between baseline and follow-up for any measure (*p* > 0.05) between the three groups (Table 5).

### 3.5. Adjustments before the Full RCT

Based on the results from this pilot and feasibility study and from the information obtained at the final workshop with the participating physiotherapists, a number of adjustments are necessary before continuing to a full RCT. To improve the recruitment rate, physiotherapists and occupational therapists at the orthopedic department will inform patients about the study and provide a short, written information sheet before the patient is discharged. Before measuring postural sway with the IMU, the procedure needs to be standardized, with clearer, written instructions provided to the physiotherapists. Postural sway also needs to be measured more frequently during the intervention period to obtain improved information on how postural sway changes over time. Moreover, in order for the physiotherapist to give feedback to the patient, information from the IMU needs to be available in time before the next planned home visit. This is to ensure that the participants receive an update on their training and mobilization. Since not reporting adherence in clinical trials can influence the therapeutic validity of the trial, adherence will also need to be measured in the full RCT [38]. After discussion within the research team and with the physiotherapists involved in the project, it was decided that the measures of the fear of falling and satisfaction with the rehabilitation were to also be included at baseline and follow-up in the full RCT. Since the exercises in the HIFE program were adapted for each individual, these HIFE exercises were quite similar to those delivered in the control group’s standard rehabilitation. Therefore, the decision to change to a two-armed RCT instead was made, with the intervention including standard rehabilitation + the IMU and the control group including standard rehabilitation only. Adjustments to the protocol will be reported in ClinTrial.gov before starting the full RCT.

## 4. Discussion

This pilot study shows that the recruitment rate was considered high, while the collection of primary data was considered high, except for postural sway, which was considered low. The collection of secondary outcomes was considered low. No harmful events were reported. The study also shows that there was no gain in terms of balance, everyday activities, and health-related quality of life after hip fracture for the group randomized to HIFE rehabilitation with or without an IMU compared to the control group. A larger RCT was found to be feasible after the adjustment of outcome data.

The postural sway measured with the IMU displayed large variations. This might be due to a number of reasons. The IMU device has shown good reliability compared to a force plate [16], but this was on a healthy, young population. To the best of our knowledge, no reliability study with an IMU device in an older population with hip fractures has been conducted. Therefore, it is difficult to assess whether this is the most optimal method to objectively measure balance for older persons with hip fractures. Another study on neurogeriatric patients showed low-to-moderate reliability for measuring postural sway with an IMU device and also questioned the clinical use of the device [39]. Moreover, the IMU device cannot differentiate between fast and slow sway, and this could be why no differences within and between groups were found. To obtain a more accurate measure of postural sway, more measures need to be carried out during the intervention period. In this study, we found a significant increase in functional balance within all three groups, with no differences between the groups. In both the HIFE program and the control group, balance exercises were delivered [40,41]. The current literature shows balance exercises to be very important in hip fracture rehabilitation to regain better balance, improvement in gait, physical function, lower-limb strength, and ADL [42]. ADL capacity after a hip fracture fluctuates from individual to individual depending on age, comorbidities, cognition etc., [43]. ADL is at its lowest post-fracture and its highest at 6 months and continues up to 1 year post-fracture. After 5 years, there is a decrease in ADL performance; this might be due to other aging-related comorbidities [43]. Compared to pre-fracture, there is a decrease in ADL [44]. A systematic review found that rehabilitation has a small-to-moderate effect on ADL in the short and long term (1 year) [13]. This review also showed a small effect of rehabilitation after hip fracture on HRQoL in the short and long term. There is a large loss in HRQoL compared to that pre-fracture, especially in older populations, and this might be explained by the loss of selfcare, usual activities, and mobility [45,46]. This could be one of the reasons why HRQoL displayed no significant change within intervention group 1, as there might have only been a small effect on HRQoL after the hip fractures. Moreover, the exercises in HIFE were found to be too difficult for our study population, and the exercises in HIFE were therefore adapted to each individual in a way that made them similar to those in the control intervention.

The recruitment rate was satisfactory (52%), but there may be several reasons for it not being higher. Those who experience hip fractures are often frail with multiple comorbidities, e.g., cognitive impairments, which may affect their ability to participate in the study [12]. Potential participants may also have been advised against participating by health care personnel or family. A lack of interest, fear, or avoidance may also affect recruitment rates [47].

The retention rate was 78%, which may be due to the individually tailored interventions. Due to loss of mobility and high age, it is important that interventions after hip fracture meet the individual needs and personal prerequisites required to enhance motivation and, therefore, improve outcome [48]. In total, seven participants dropped out, and out of these, two dropped out due to readmission to hospital. The ability to collect primary and secondary data at baseline was considered high for all outcomes, except for the postural sway measurement with the IMU. The reason for the difficulty in collecting data on postural sway might be due to the problems with the IMU device or to the standardization of the instructions.

The decision that a full RCT is feasible to perform was made after discussions in the team concerning the possibilities of adjusting the protocol in a way that recruitment and abilities to collect outcomes at follow-up could be more efficient. In particular, the decision to collect data on postural sway more frequently, together with position and mobility data from the body-worn IMU, may improve the ability to truly individualize rehabilitation.

### Strengths and Limitations

Firstly, there were some problems when collecting data on postural sway with the IMU device. The device did not show when it was fully charged, and, therefore, it sometimes turned out to be uncharged at the time of data collection. Moreover, the instructions for standardizing the measuring procedure were not clear enough, which limited our results on postural sway. Moreover, due to the lack of data on compliance rate, we did not know if the intervention groups received more training sessions than the control group or the compliance rate. We did not have information regarding the pre-fracture function of the participants.

In this study, we lack qualitative information about how the participants and the PT perceived the intervention. The User Satisfaction Evaluation Questionnaire (USEC) is designed to be answered by a user of technology. In our study, the physiotherapists were the users, but the final workshop with the physiotherapists revealed that they considered USEC to be irrelevant, and, thus, they had not filled it in. Therefore, there were no data from USEC to analyze. We consider the RCT design and the individualized but controlled exercise program as strengths of this study since the program was easy to adjust to each individual’s progression.

## 5. Conclusions

The results of this pilot and feasibility study show that a full RCT is feasible and can be conducted after adjustments to the protocol. The adjustments include changing to a two-armed RCT, adding two measurements and excluding one, and modifying the instructions and handling of the IMU when measuring postural sway. The result also shows no gain in terms of balance, everyday activities, and health-related quality of life after hip fracture for the group randomized to the HIFE rehabilitation with or without an IMU compared to the control group.

## Figures and Tables

**Figure 1 ijerph-20-03107-f001:**
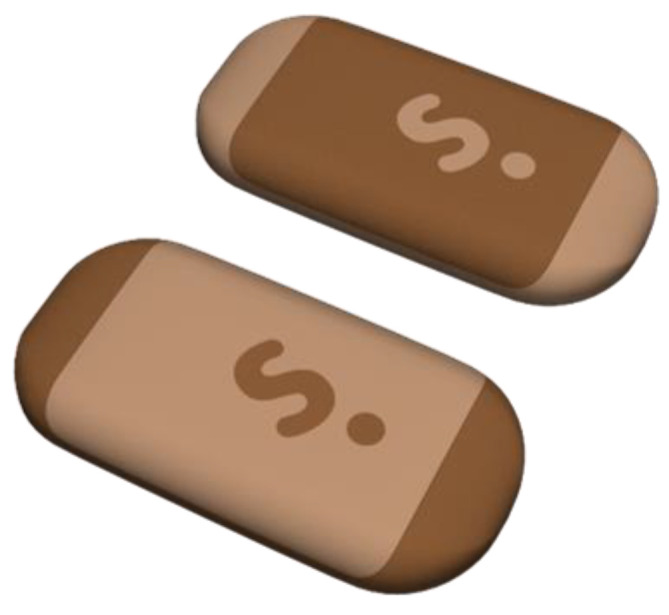
The inertial measurement unit used in intervention group 2.

**Figure 2 ijerph-20-03107-f002:**
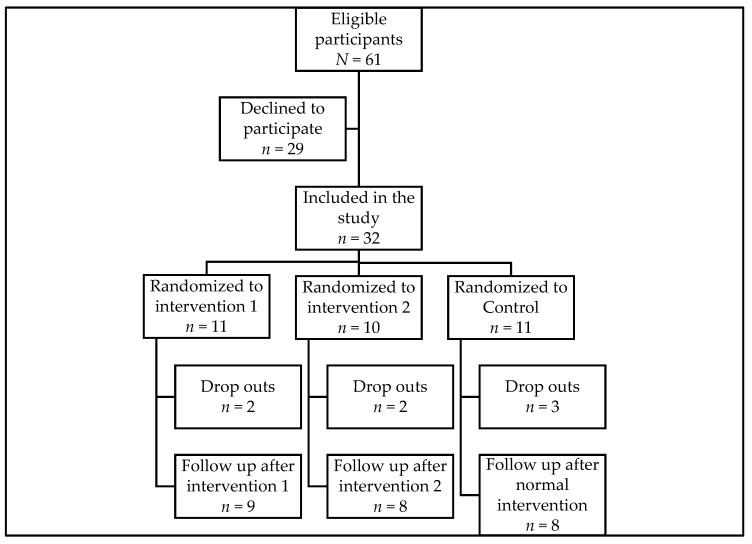
Flow chart of the study. Reasons for drop out were not known (*n =* 5) and admission to hospital (*n =* 2).

**Table 1 ijerph-20-03107-t001:** Categories and examples of HIFE exercises.

Categories	Examples of Exercises
A: Static and dynamic exercises in combination with lower-limb strength exercises	Squats, body weight transfer, sit to stand, lunges, step-up, stair walking
B: Dynamic balance exercises in walking	Walking exercises with various difficulty levels, step over, change of walking surface
C: Static and dynamic balance exercises in standing	Weight transfers, turning head in different directions, reaching for an object, catching and throwing a ball, kicking a ball
D: Lower-limb strength exercises with continuous balance support	Squats, sit to stand, heel raises, weight transfer sideways, stair walking
E: Walking with continuous balance support	Walking in different directions, walking and turning

**Table 2 ijerph-20-03107-t002:** Intensity scales for HIFE exercises.

Training Session	High Intensity	Medium Intensity	Low Intensity
Strength training exercises	Sets of 8–12 RM	Sets of 13–15 RM	Sets of >15 RM
Balance training exercises	Postural stability fully challenged *	Postural stability not fully challenged or fully challenged in a minority of the exercises	Postural stability not challenged

* Exercise performed close to the limit of postural stability.

**Table 3 ijerph-20-03107-t003:** Median and interquartile range (IQR) for baseline measures for the total group and the total group divided into intervention groups 1 and 2 and the control group.

Measure	Total Sample *n =* 32 Median (IQR)	Intervention 1 *n =* 11 Median (IQR)	Intervention 2 *n =* 10 Median (IQR)	Control *n =* 11 Median (IQR)
Women/men	25/7	10/1	7/3	8/3
Age (years)	85 (12)	83 (13)	86 (16)	85 (16)
Intervention days	77 (38)	58 (33)	77.5 (26)	80.5 (19)
FBG *	6 (3)	7 (3)	5,5 (2)	8 (2)
BI	62.5 (24)	60 (20)	55 (18)	75 (20)
EQ5D index	0.508 (0.211)	0.465 (0.226)	0.487 (0.382)	0.556 (0.110)
EQ5D VAS in mm	50 (30)	40 (45)	44 (43)	50 (25)
PSOEAP mm/s	15 (22)	15 (38.3)	17 (13.5)	7 (19)
PSOEML mm/s	4 (4)	4.5 (10.3)	5.0 (2.5)	3.0 (2.3)
PSCEAP mm/s	10 (19.5)	16.5 (38.5)	16 (16)	9.5 (8.8)
PSCEML mm/s	5 (8)	5 (18)	5 (4)	7 (7.5)

* FBG, Functional Balance for Geriatric Patients; BI, Barthel index; EQ5D index, EuroQol 5-Dimension index; EQ5D VAS, EuroQol 5-Dimension Visual Analogue Scale; PSOEAL, postural sway open eyes anterior/posterior; PSOEML, postural sway open eyes medial/lateral; PSCEAP, postural sway closed eyes anterior/posterior; PSCEML, postural sway closed eyes medial/later.

**Table 4 ijerph-20-03107-t004:** On-treatment analysis of differences in change between baseline and follow-up within groups.

	Within-Group Differences, Intervention 1 (*n =* 9)			Within-Group Differences, Intervention 2 (*n =* 8)			Within-Group Differences, Control (*n =* 8)		
Measure	Median Change	CI	*p*-Value	Median Change	CI	*p*-Value	Median Change	CI	*p*-Value
FBG	5	4–7	**0.007**	7.5	2–13	**0.018**	5.5	4–13	**0.018**
BI	20	5–40	**0.008**	35	10–40	**0.011**	22.5	15–35	**0.012**
EQ5D index	0.185	0.092–0.457	**0.011**	0.228	0.165–0.339	**0.012**	0.193	0.068–0.253	**0.018**
EQ5D VAS	20	−5–35	0.262	26	5–45	**0.036**	30	15–45	**0.018**
PSOEAP mm/s	12	−6–27	0.225	8	3–13	0.068	1	0–13	0.102
PSOEML mm/s	−1	−12–12	0.581	0	−10–3	0.705	2	−5–7	0.465
PSCEAP mm/s	7.5	−7–20	0.705	−6	−22–2	0.144	0	−11–3	1.000
PSCEML mm/s	0.5	−4- 14	0.465	0	−18–5	0.854	1	−1–3	0.414

CI, 95% confidence interval for median change; *p*-value is difference in median change, calculated using the Mann–Whitney U test; FBG, Functional Balance for Geriatric Patients; BI, Barthel index; EQ5D index, EuroQol 5-Dimension index; EQ5D VAS, EuroQol 5-Dimension Visual Analogue Scale; PSOEAL, postural sway open eyes anterior/posterior; PSOEML, postural sway open eyes medial/lateral; PSCEAP, postural sway closed eyes anterior/posterior; PSCEML, postural sway closed eyes medial/lateral. Statistically significant *p*-values are shown in bold.

**Table 5 ijerph-20-03107-t005:** On-treatment analysis of differences in changes between baseline and follow-up between groups. Median change and CI between intervention 1 and intervention 2, intervention 1 and control, and intervention 2 and control.

	Between-Group Differences, Intervention 1 and Intervention 2 (*n =* 17)		Between-Group Differences, Intervention 1 and Control (*n =* 17)		Between-Group Differences, Intervention 2 and Control (*n =* 16)		*p*-Value for Median Difference in Change
Measure	Median Change	CI	Median Change	CI	Median Change	CI	
FBG	5	4–10	5	5–6	5	4–10	0.943
BI	30	15–40	20	15–35	30	20–35	0.283
EQ5D index	0.191	0.137–0.339	0.188	0.104–0.253	0.210	0.123–0.256	0.630
EQ5D VAS	25	0–30	30	15–35	30	20–45	0.322
PSOEAP mm/s	9	−1–13	6	−1–13	7	0–13	0.921
PSOEML mm/s	−2	−10–3	0	−5–7	2.5	−5–3	0.724
PSCEAP mm/s	−2.5	−8–16	1	−7–16	−2	−11–2	0.304
PSCEML mm/s	0	−4–5	1	−4–3	1	−18–3	0.920

CI, 95% confidence interval for median change; *p*-value is difference in median change, calculated using the Kruskal–Wallis test; FBG, Functional Balance for Geriatric Patients; BI, Barthel index; EQ5D index, EuroQol 5-Dimension index; EQ5D VAS, EuroQol 5-Dimension Visual Analogue Scale; PSOEAL, postural sway open eyes anterior/posterior; PSOEML, postural sway open eyes medial/lateral; PSCEAP, postural sway closed eyes anterior/posterior; PSCEML, postural sway closed eyes medial/lateral.

## Data Availability

The data used in this study contain sensitive information about the study participants, and they did not provide consent for public data sharing. The current approval by the Swedish Ethical Review Authority does not include data sharing. A minimal data set could be shared on request from a qualified academic investigator for the sole purpose of replicating the present study, provided that the data transfer is in agreement with EU legislation on the general data protection regulation and approval by the Swedish Ethical Review Authority.

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
