# Peer review of "Rehabilitation Outcomes Following Hip Fracture of Home-Based Exercise Interventions Using a Wearable Device—A Randomized Controlled Pilot and Feasibility Study"

_ijerph, 2023, doi:10.3390/ijerph20043107_

Round 1

Reviewer 1 Report

This is a small 3 arm RCT of home-based exercise for individuals after hip fracture with effect and feasibility research questions. Hip fracture rehabilitation is an important area of study.

Introduction:

The literature cited on the effects of rehabilitation after hip fracture should be further developed. There are a number of recent systematic reviews and meta-analyses and known effective interventions in this area that are not represented by the synthesis provided. The authors state that “Rehabilitation after a hip fracture should conform to the patient’s needs and level of mobility, and rehabilitation plans should focus on balance, functional activities, endurance and progressive strength training for the lower limbs.” The authors should provide a synthesis of the current best evidence specific to the research question to support their statements.

The manuscript is limiter be the lack of a conceptual or theoretical framework for the mechanism of action of the intervention, or feasibility or outcome measurement.

After stating that there is “insufficient evidence to determine which type of mobilization or program is most beneficial following a hip fracture” the authors chose balance, progressive strength training, and functional exercises without specifying the evidence to support them. It is not clear from the introduction what specific gap the study is designed to address.

Different IMUs, different models of the same IMU, and different data management algorithms have different measurement characteristics, especially in older adults. Therefore, specific detail on the IMU, model and year should be reported, and the evidence cited should be specific to it and the population. The authors mention changes to features of the IMU for the study. Any study-specific changes to the IMU should be accompanied by reliability and validity testing.

There is insufficient rationale provided for postural sway as measured by an IMU as the primary outcome for this study. Notwithstanding the statement that postural sway is an indicator of fall risk, it is not in line with primary goals of rehabilitation or the evidence in hip fracture rehabilitation. It is also not clear why this study is described as a pilot study if it was powered to answer its main research question.

Section 2.7 Statistical methods

The description of the power analysis and sample size calculation did not specify the outcome and effect size on which the sample size was based. In addition, for each measure, the minimal clinically important difference and/or minimal detectable change should be provided if available.

The interpretation may be misleading. “This pilot study showed that, regardless of rehabilitation model, all 3 groups improved in functional balance, self-rated activity of daily living and HRQoL” indirectly implies that all three treatments were effective. This study was designed to test differences between the three groups and is not able to test against temporal effects without treatment. Therefore, the interpretation should be improved by making it more specific and better aligned with the hypotheses tested, and less about the within groups differences over time.

The description of the retention rate seems out of line with usual interpretation of risks for bias. A 22% loss to follow up would not usually be interpreted as small.

Author Response

Thank you for your time and effort to help us improve our manuscript. Please find our response and actions in the attached file. 

Reviewer 2 Report

General minor revision

abbreviations are not written out at first mention (see abstract IMU). please check at all places with abbreviations if they correspond to the criteria in publications.

Major revision:

Do you measure the fraily status of your patients before and after surgery? if so please explain your assessment and put it into account. If not please explain why you don't measure such an important and relevant issue in rehabilitation management.

In the pilot study, were patients offered additional prevention programs to protect patients from falls or other complications and to support patients in their home-base?

It does not seem right to me in a pilot and feasibility study to assume improved outcomes in all groups, regardless of the rehabilitation model, because due to the pilot nature the study does not have the power to make such a statement with certainty. Please correct your statement. 

Concerning Methods:

Please explain how were home exercise interventions monitored in the control group when no IMU was available?

Recruitment: the rate of rectrution is very low. Please explain how the increase in recruitment rate in the RCT will be achieved? Which recruitment strategy will you use? Do you see potential for optimization in recruitment within your structures?

Limitations: please explain how you plan to put the problems of collecting data into perspective with the IMU device? what additional device could be available so that you measure validated data in the RCT with little or no problems alternatively to IMU device?

Author Response

(The authors gave the same response as above.)

Reviewer 3 Report

Thank you for submitting this article "Rehabilitation outcomes following hip fracture in home-based exercise interventions using a wearable device - a randomized controlled pilot and feasibility study". It is good to see research taking the time to publish protocols and feasibility trials prior to deciding whether to progress the research further. I note that this work is a follow up to a previously published protocol, and therefore I think it is relevant to publish its associated results. The manuscript is well-written. 

However, I have some concerns regarding the protocol, but mostly on how the authors have come to the conclusion that a full trial is warranted based on these results. Id like to confirm that I am not saying that it isnt warranted, but for me there is a lack of rationale or detail that helps make that conclusion strong. I therefore believe that there are major changes required to this manuscript in order to strengthen its conclusions. I have highlighted some of my questions or concerns in each section below, acknowledging that it wont be possible to alter some of the items now. Nonetheless, I think there may need to be a greater highlighting of limitations or a longer list of changes that need to be made to the protocol if advancing further. The aim of this is not to be negative but to be honest to readers regarding how changes and decisions to progress or not are made. 

Abstract:

Some additional information or clarifications would be useful here. Please note how many people took place in this pilot RCT. Please also spell out the abbreviated words in full in the abstract the first time they are written. It would also be useful to add another sentence to the conclusion regarding why you determine that a full RCT can be completed with protocol changes. 

Introduction

I would like more information about what generally constitutes rehab and whether there are any guidelines already in place regarding what should be included. You have mentioned that it is unclear what is most beneficial but I would be surprised if there is not general evidence or aims as to what rehabilitation should aim to do and work. Similarly, what evidence exists for the HIFE programme already, and how does it differ from what you refer to standard rehabilitation. I think there needs to be a lot more clarity around this. Additionally, I think there needs to be further information given regarding the wearables. Has it been implemented in this population before? Are there concerns regarding its feasibility in this population and if so what are they? 

Finally, there are multiple guidelines that exist regarding the need to do pilot/feasibility testing prior to conducting full RCTs. It would be useful to make reference to these and outline some of the headings that need to be considered within these guidelines to rationalise why you are using them here.

I see the protocol for this has been published already. In my opinion some of the parts I have highlighted above are listed in this protocol. I would suggest firstly highlighting within the introduction that this protocol has been published, rather than waiting until the methods, and use the protocol as a basis for how you set this paper up. Reference it earlier, highlight how you have previously reported on the need to explore and test this and why and then focus on the importance of the pilot/feasibility test. 

Methods 

With regards to the wearable I think it would be beneficial to put the information about the IMU into the intervention section so that it is clear what was involved in deploying this. I would also include an image of the IMU. If the feedback was only provided by the PT, then how often was this feedback provided, how was it provided and was any fidelity assessment completed to ensure that they actually provided it? When describing the control group it really needs to be clearer how this is different to the intervention and what the control group contained. For a feasibility/pilot it needs to be clearly documented what took place in the sessions in this arm. At the moment I dont believe someone would be able to replicate the control group with the detail provided. Please consider it as a supplemental file if word count is tight. 

With regards to your outcome measures, in the context of a pilot/feasibility trial, should the primary outcomes not be related to the running of the trial, rather than patient outcomes. Patient outcomes are generally only considered as exploratory in trials such as this, and so I'm interested in why they are the primary measures in this context? 

Im also interested in why you have not included any patient facing feasibility outcomes either? For example, were they satisfied with wearing the wearable and the fact that they dont appear to have received direct, live feedback from it? Was it comfortable for them? Did it add additional time to the work of the PT? Did the PTs have any issue in interacting with it or deploying it? I appreciate that your protocol has been published but these appear to be potentially important omissions when it comes to feasibility. 

Finally for the methods, I have to assume that the power calculations were correct but the proposed numbers for a fully powered RCT appear small. Indeed, you've remarked yourself that the sample size is small. Regardless of the size of the sample however, are they normally distributed or not? If they are I would suggest that parametric tests would be appropriate? Also, did you adjust the analysis for the number of comparison tests undertaken (e.g. Bonferroni or similar)

Results 

Figure 1 needs to include reasons for drop-outs or loss to follow up

Discussion 

I think it needs to be clearer how you determined whether it is worth continuing to go towards a RCT. What criteria of success needed to be reached in terms of recruitment rates, retention rates etc. Was this agreed a-priori, have you used previous guidance? 

For me this is especially important given some of the findings. You remarked that postural sway with an IMU was validated in healthy adults, rather than this population. Can we really justify continuing with this device or measure in a full trial given this knowledge? I also would suggest that there are patient specific details that are important to consider also. What was the baseline mobility of the patients, were they using mobility aids, were they living independently or not? This will be important for an RCT. Similarly, you have hypothesised why people may not have taken part but this is important information to capture in a feasibility trial. I appreciate you may not always receive responses from people, but any data on this element is helpful at this stage of a trial and to not have collected should be considered a limitation. 

For me there needs to be a much more critical reflection of how the suggestion that a fully powered RCT should be conducted is needed. 

Minor changes:

Your initial sentence regarding greater life expectancy creating more fractures feels like too generalised a start. I would suggest removing this. 

Please spell your abbreviated words in full the first time they are listed, for example IMU

Author Response

(The authors gave the same response as above.)

Round 2

Reviewer 1 Report

The manuscript has been improved by the revisions. The Clinical Trials Registry number provided points to a full RCT, not a feasibility pilot. 

The lack of a theoretical framework for the role of postural sway relative to falls limits the impact of this work. Since falls are known to be multifactorial, explicitly addressing the role of postural control as measured by sway is critical. For example, the citations for postural sway predicting falls should indicate the strength of the relationship - how predictive? 

This study did not measure fidelity of the intervention, which is an important potential alternative explanation for null findings. The initial intervention was developed for older adults in nursing homes; was it changed for a home-based population? 

Line 131 states that IMU's are reliable and valid. This statement along with the prior reporting on the IMU raises important questions about the work. As indicated in my prior comments, it is well-documented that the reliability and validity of IMU's varies by the make, model, any algorithms used, the population and the application. The authors should indicate the reliability estimates (rather than stating "excellent") for the device/algorithm used in the study. 

Author Response

Thank you for helping us improve our manuscript. Please see the attachment.

Reviewer 2 Report

Dear sir or madam,

thank you for the revision of your manuscript. I have no further comments. 

Kind regards 

Yürek

Author Response

Thank you for helping us improve our manuscript. Please see the attachment

Reviewer 3 Report

Thank you for addressing my comments.

I personally would still be a little reluctant to say there is enough evidence here to suggest a fully powered trial, but I am more satisfied that the conclusions made are now based on an appropriate assessment of feasibility.   I also personally feel that patient reported outcomes regarding comfort/satisfaction/feasibility should always be included in feasibility studies, and would strongly encourage you to include these as standard in future studies. They should always be within the aim as the research cannot be successful without their engagement. 

Nonetheless, you have responded to my queries in full, thank you. 

Author Response

(The authors gave the same response as above.)
